# HEPATOKIN1 is a biochemistry-based model of liver metabolism for applications in medicine and pharmacology

Nikolaus Berndt[1], Sascha Bulik[1,2], Iwona Wallach[1], Tilo Wünsch[3], Matthias König [4], Martin Stockmann[2], David Meierhofer [5] & Hermann-Georg Holzhütter[1]

The epidemic increase of non-alcoholic fatty liver diseases (NAFLD) requires a deeper understanding of the regulatory circuits controlling the response of liver metabolism to nutritional challenges, medical drugs, and genetic enzyme variants. As in vivo studies of human liver metabolism are encumbered with serious ethical and technical issues, we developed a comprehensive biochemistry-based kinetic model of the central liver metabolism including the regulation of enzyme activities by their reactants, allosteric effectors, and hormone-dependent phosphorylation. The utility of the model for basic research and applications in medicine and pharmacology is illustrated by simulating diurnal variations of the metabolic state of the liver at various perturbations caused by nutritional challenges (alcohol), drugs (valproate), and inherited enzyme disorders (galactosemia). Using proteomics data to scale maximal enzyme activities, the model is used to highlight differences in the metabolic functions of normal hepatocytes and malignant liver cells (adenoma and hepatocellular carcinoma).

[1] Institute of Biochemistry Computational Systems Biochemistry Group, Charité – Universitätsmedizin Berlin, Charitéplatz, 110117 Berlin, Germany. [2] German Federal Institute for Risk Assessment Max-Dohrn-Straße 8-10, 10589 Berlin, Germany. [3] Department of General, Visceral and Transplantation Surgery Augustenburger Platz, Charité – Universitätsmedizin Berlin - Campus Virchow-Klinikum, 113353 Berlin, Germany. [4] Institute for Biology, Institute for Theoretical Biology, Humboldt-University Berlin, Invalidenstraße 43, Haus, 410115 Berlin, Germany. [5] Max Planck Institute of Molecular Genetics/Mass Spectroscopy, Ihnestraße 63-73, 14195 Berlin, Germany. Correspondence and requests for materials should be addressed to H.-G.H. (email: hergo@charite.de)

nferring the response of a biological system to external or internal perturbations from the properties and interactions of its constituting molecules is a central goal of systems biology[1]. With regard to metabolic systems, reaching this goal requires the establishment of mathematical models enabling the computation of metabolite concentrations and fluxes at given external conditions (nutrients and hormones), gene expression level of metabolic enzymes, and the systems history (e.g., current filling of nutrient stores).

Chemical reactions and mass transport are the basic processes in a metabolic network. They are catalyzed by specific enzymes and transport proteins that are regulated in multiple ways by their immediate ligands, allosteric effectors, hormone-dependent reversible phosphorylation, and variable gene expression. Often, a single specific regulatory enzyme feature is key for the regulation of a complete metabolic pathway. For example, the strongest regulator of the mitochondrial fatty acid transporter, carnitine palmitoyltransferase (CPT), is the competitive inhibitor malonyl-CoA. Decrease of malonyl-CoA concentration during the overnight fast is lifesaving because activation of CPT enables the enhanced oxidation of fatty acids to acetyl CoA and hence the formation of glucose-sparing ketone bodies in the liver[2]. This example underlines the importance of biochemistry-based kinetic models that incorporate such important regulatory features of enzymes.

The strong medical interest in a better understanding of the molecular processes underlying the regulation of liver metabolism arises from the fact that an ongoing metabolic imbalance of the organ, e.g., due to excessive intake of drugs, alcohol or fructose, may result in an abnormal accumulation of lipids (steatosis) thereby increasing the risk of developing serious liver diseases such as hepatitis, cirrhosis, and cancer[3]. Aiming at the in vivo assessment of liver metabolism, we developed a kinetic multi-pathway model of hepatocytes with hitherto unprecedented scope and level of detail. The model includes the regulation of enzyme activities by allosteric effectors, hormone-dependent reversible phosphorylation, and variable protein abundances. For each enzyme, rate equations have been developed that take into account the enzyme's kinetic and regulatory features as revealed and quantified by means of in vitro assays. In the following, we give an overview of the model while referring the interested reader to the extensive Supplementary material containing all technical details. We focus in the main text on simulations of the dynamic metabolic output of the liver at different plasma profiles of metabolites and hormones. Using quantitative proteomics data for the scaling of maximal enzyme activities, the model opens the goal for a quantitative functional interpretation of gene expression changes. We applied this approach to reveal the patient-specific metabolic profile of adenoma and HCC. In summary, our model provides a powerful tool for computational studies of liver metabolism in health and disease.

## Results

**Model description.** The metabolic part of the kinetic model comprises the major cellular metabolic pathways of cellular carbohydrate, lipid, and amino acid metabolism of hepatocytes (see Fig. 1). The model also contains key electrophysiological processes at the inner mitochondrial membrane, including the membrane transport of various ions, the mitochondrial membrane potential, and the generation and utilization of the proton-motive force. The time-dependent variations of model variables (=concentration of metabolites and ions) are governed by first-order differential equations. Time-variations of small ions were modeled by kinetic equations of the Goldman–Hodgkin–Katz

type as used in our previous work[4]. The rate laws for enzymes and membrane transporters were either taken from the literature or constructed on the basis of published experimental data. The mathematical form of the kinetic rate laws for enzymes and membrane transporters was dictated by the reaction mechanism and taken from enzymological in vitro studies, preferably for the liver of the rat or, and if not available, in the order mice → human → bovine → dog. The same ranking of species was applied for the retrieval of numerical values for the kinetic parameters. For every kinetic parameter we cite one experimental reference. If several numerical values for the same kinetic parameter and the same species were reported, we used one representative value that fits with the majority of the reported values and that—whenever possible—was taken from an enzyme assay that reported consistent values for other kinetic parameters. Mathematical terms in the rate law related to allosteric enzyme effectors, which are not included in the model were neglected. Their average contribution is indirectly contained in the fitted $V_{max}$ values. More than 90% of the parametric model input holds for the rat liver. Supplementary Note 1 contains all kinetic equations and model parameters sorted by individual pathways. All model simulations were performed using MATLAB, Release R2011b, The MathWorks, Inc., Natick, Massachusetts, United States.

The regulation of key reaction steps in mutually opposing pathways (e.g., glycolysis and gluconeogenesis, lipid synthesis, and lipolysis) by hormone-sensitive reversible enzyme phosphorylation represents an important regulatory principle to control the direction of the net flux[5]. The signaling part of the model comprises the insulin and glucagon dependent regulation of key regulatory enzymes by reversible phosphorylation. The rate laws for these enzymes take into account that the phosphorylated and de-phosphorylated states of the enzyme possess differing maximal activities and kinetic properties. As in our previous work[6], we used phenomenological mathematical functions to relate the enzyme's phosphorylation state to the plasma concentrations of glucose (Supplementary Note 2)

The model boundaries are given by the metabolite and hormone concentrations in the extracellular space. The flux through reactions not considered in the model were put to zero, i.e., carbon influx into and efflux from the modeled metabolic subsystem is exclusively mediated by the exchange fluxes through the plasma membrane.

**Model calibration.** Except for the $V_{max}$ values, which may vary owing to variable gene expression, the numerical values for all other model parameters were taken from reported kinetic studies of the isolated enzyme. The $V_{max}$ values (Supplementary Table 1) were estimated by fitting the model to 585 measurements of exchange fluxes and internal metabolites obtained in 25 different experiments carried out with perfused livers or isolated hepatocytes, covering a broad range of 21 important liver functions (Supplementary Note 3). A description of each simulated experiment, its physiological relevance, the initial conditions of the simulation, and a comparative plot of measured and simulated data is given in the Supplementary Note 3. For each of the 585 model simulations, we forced the concentration values of 170 internal metabolites to remain within the concentration ranges defined on the basis of in vivo and in vitro measurements (see Supplementary Data 1 and Supplementary Fig. 1). For more details of the parameter fitting procedure see Methods.

An illustrative example of a calibration simulation is shown in Fig. 2 depicting the relationship between the partial oxygen pressure (pO2) and selected model variables. Under normoxic conditions, ATP is almost exclusively generated by oxidative

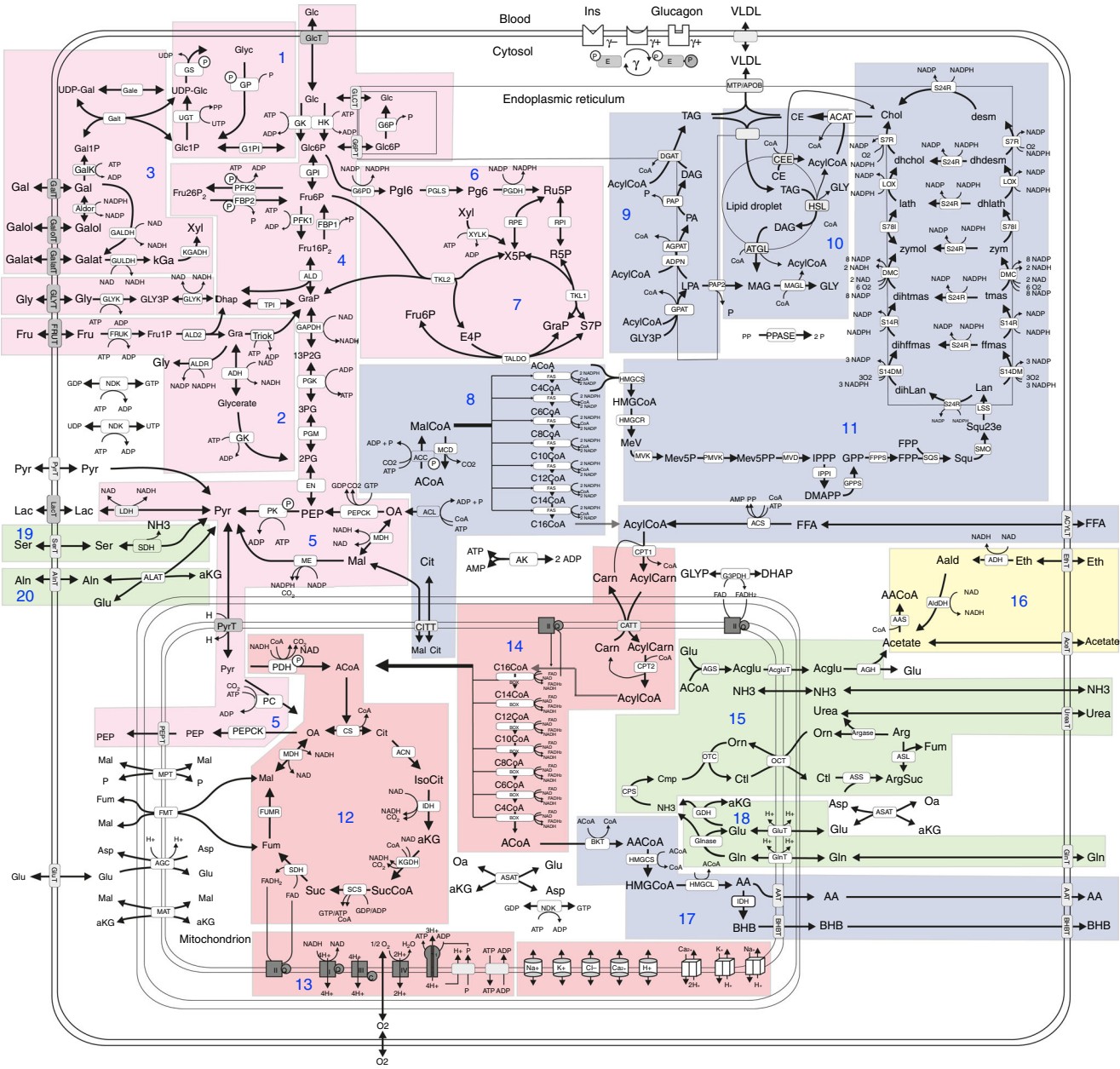

**Fig. 1** Reaction scheme of the metabolic sub-model. Reactions and transport processes between compartments are symbolized by arrows. Single pathways as defined in biochemical text books are numbered and highlighted by different coloring: (1) glycogen metabolism, (2) fructose metabolism, (3) galactose metabolism, (4) glycolysis, (5) gluconeogenesis, (6) oxidative pentose phosphate pathway, (7) non-oxidative pentose phosphate pathway, (8) fatty acid synthesis, (9) triglyceride synthesis, (10) synthesis and degradation of lipid droplets and synthesis of VLDL lipoprotein, (11) cholesterol synthesis, (12) tricarbonic acid (TCA) cycle, (13) respiratory chain & oxidative phosphorylation, (14) β-oxidation of fatty acids, (15) urea cycle, (16) ethanol metabolism, (17) ketone body synthesis, (18) ammonia formation, (19) serine utilization, and (20) alanine utilization. Small cylinders and cubes symbolize ion channels and ion transporters. Double-arrows indicate reversible reactions, which may proceed in both directions according to the value of the thermodynamic equilibrium constant and cellular concentrations of their reactants. Reactions are labeled by the short names of the catalyzing enzyme or membrane transporter given in the small boxes attached to the reactions arrow. Metabolites are denoted by their short names. Full names and kinetic rate laws of reaction rates are outlined in Supplementary Note 1. The full names of metabolites and a comparison of experimentally determined and calculated cellular metabolite concentrations is given in Supplementary Data 1

phosphorylation. If the external O₂ level drops below critical levels of about 15 mmHg, the cellular O₂ level becomes too low to saturate complex IV of the respiratory chain, which decreases the flow of electrons, the proton-motive force, and thus the rate of oxidative phosphorylation. The fall of ATP blocks ATP-dependent reaction steps in anabolic metabolic pathways, such as gluconeogenesis and urea synthesis. Therefore, these two cardinal anabolic functions of the liver become severely restricted.

Inspection of the transition from the normoxic to the hypoxic state elicited by a sudden drop of oxygen allows to trace back the causal chain of molecular events underlying the relationship between falling O₂ levels and reduced rates of hepatic glucose and

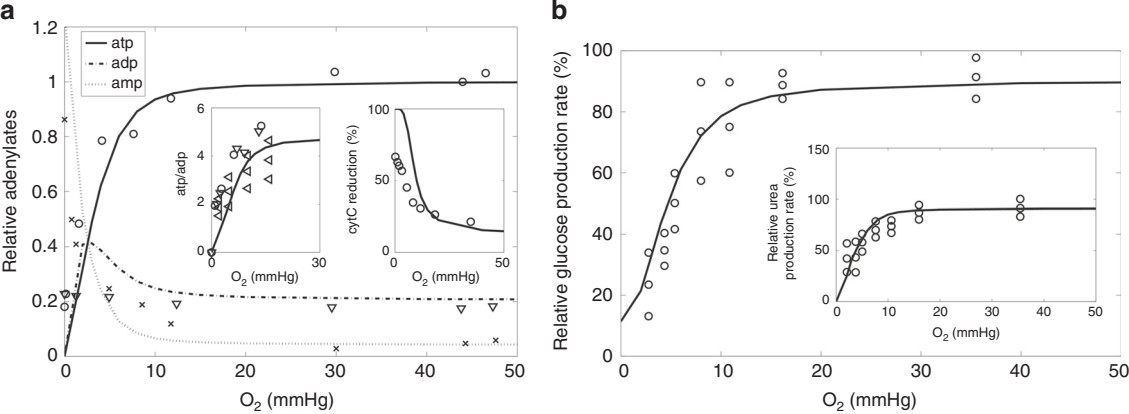

**Fig. 2** Effect of hypoxia on selected model variables. **a** Adenine nucleotides ATP, ADP, and AMP. Small insertions: ATP/ADP ratio and reductions state of cytochrome c. **b** Relative production rates of glucose and urea (small insertion). Discrete symbols refer to experimental data from various publications. For details see Supplementary Note 3

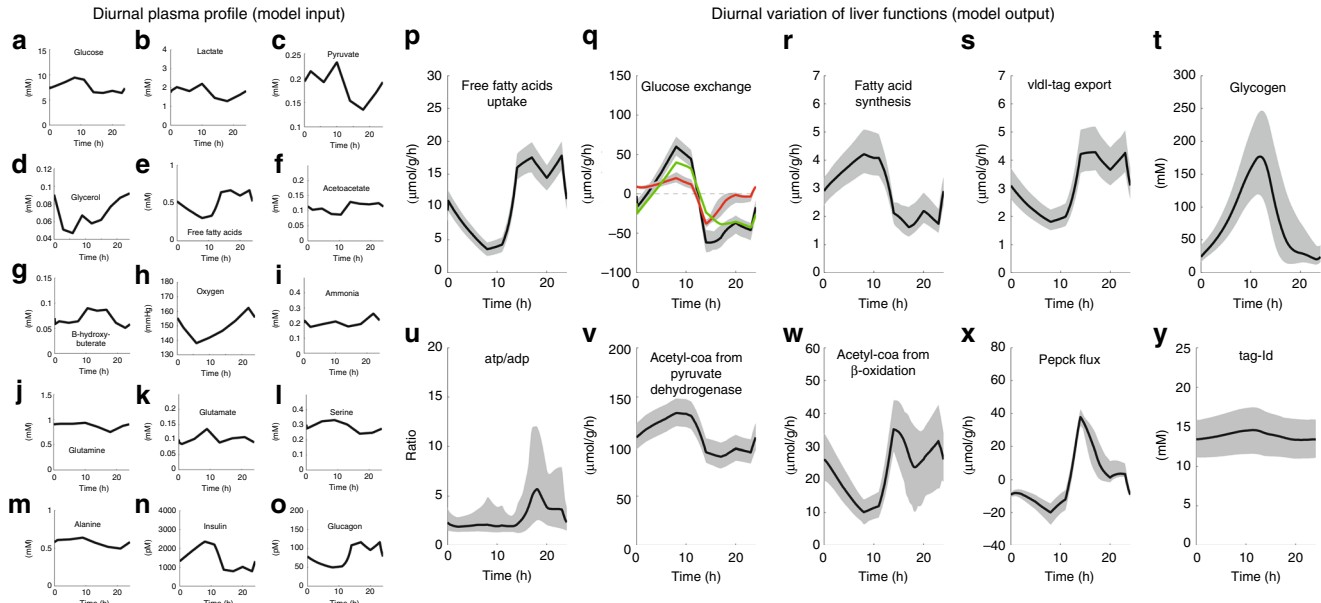

**Fig. 3** Simulated diurnal changes of selected metabolic functions of a normal rat liver. Left panels: 24 h plasma profile of hormones and metabolites of an ad libitum fed rat serving as model input. **a** glucose, **b** lactate, **c** pyruvate, **d** glycerol, **e** free fatty acids, **f** acetoacetate, **g** β-hydroxy butyrate, **h** oxygen, **i** ammonia, **j** glutamine, **k** glutamate, **l** serine, **m** alanine, **n** insulin, **o** glucagon. Right panels: Simulated diurnal profile of 10 selected metabolic functions (=model output). **p** uptake of free fatty acids, **q** glucose exchange, **r** fatty acid synthesis, **s** triglyceride export by VLDL, **t** glycogen content, **u** ATP/ADP ratio, **v** rate of pyruvate dehydrogenase (=entry of glucose carbons into the citric acid cycle), **w** rate of β-oxidation (=entry of fatty acid carbons into the citric acid cycle), **x** rate of PEPCK, **y** triglyceride content. The green and red curves in **q** represent the rate of glycolysis/gluconeogenesis and glycogen synthesis/degradation (=flux through the phosphoglucomutase). Note that negative exchange fluxes indicate net release from the liver to the plasma. Gray areas around the solid curves encompass 95% of values obtained in 100 recurrent simulations where the $V_{max}$ value of each enzyme was randomly sampled from an interval between 90 and 110% of the reference value

urea production. The experimentally observed deviation in the cytochrome c reduction state from the expected thermodynamic equilibrium at low $O_2$ levels points to electron acceptors outside the respiratory chain not considered by the model[7].

**Assessment of liver metabolism in vivo.** The metabolic state of the liver is mainly dictated by the plasma concentrations of metabolites and hormones, which are continuously changing depending on various factors such as the individual time regime of eating and fasting, the amount and composition of the food, physical activity or presence of systemic diseases like diabetes.

Our model offers the possibility to simulate these scenarios in a quantitative manner by using the plasma profile of metabolites and hormones as model input. Fig. 3. shows the simulated diurnal variation of some key metabolic functions for a normal liver (see Supplementary Fig. 2 for a larger set of 24 functions). The effect of inter-individual variability of protein abundances on the shape of the simulated trajectories was taken into account by Monte–Carlo sampling, where the $V_{max}$ values of all enzymes were randomly varied by ±10% in correspondence with reported average inter-individual variation of 19% of the liver proteome[8].

During the day and free access to nutrients, the liver takes up glucose (Fig. 3q), and stores it in glycogen (peaking around 10 h)

(Fig. 3t). Removal of excess carbohydrates from the plasma is a vital liver function preventing violent oscillations in circulating glucose after feeding. If, for example, 100 g of glucose were delivered in its entirety into the extracellular fluid, plasma glucose levels would rise by 37 mmol/l, whereas the measured rise is rarely more than 10% of this[9]. During the night, i.e., without intake of carbohydrates but ongoing glucose consumption (predominantly by the brain), the liver stabilizes the plasma glucose level by the export of glucose. Glucose release is accompanied by a significant drop of the hepatic glycogen reserve by about 80% and an increase of the rate of gluconeogenesis (cf. red and green curves in Fig. 3q)[10]. The hepatic uptake rate of fatty acids follows the plasma level of fatty acids (Fig. 3p and Fig. 3e)[11]. Owing to the inhibitory effect of insulin on Apo-B, the export rate of triglycerides to extra-hepatic organs contained in very low density lipoprotein (VLDL) is low if the glucose and insulin plasma levels are high and vice versa[12].

Of notice, endogenous de novo synthesis of fatty acids is inversely regulated against the import of fatty acids and the export of VLDL (Fig. 3r, p, s). This is mainly accomplished by regulation of the acetyl-CoA carboxylase by allosteric effectors (e.g., fatty acids) and reversible phosphorylation[13]. Contrary to the fast glycogen stores, hepatic triglyceride stores serve as long-term fuel reserve and are almost unaffected during a normal diurnal feeding cycle (Fig. 3y)[14].

Notably, the model allows to monitor intracellular metabolic changes, which are hardly accessible to direct experimentation. For example, the model predicts diurnal changes of the ATP/ADP ratio by a factor of about two. The relative share of glucose and fatty acids in the oxidative ATP production of the liver (quantified by the ratio of fluxes through the pyruvate dehydrogenase and β-oxidation both yielding Acetyl-CoA) varies between 13:1 (at 10 h) and 2:1 (at 15 h), reflecting the adaptation in fuel preference over the day. Counterintuitively, the energetically most comfortable situation with a ratio ATP:ADP ≈ 6 is reached when the uptake of fatty acids and their relative share in oxidative ATP production is highest corresponding to times when the energetic demand for VLDL synthesis and gluconeogenesis peaks. Importantly, the 24 h concentration changes of all internal metabolites remained within experimentally overserved concentration ranges (see Supplementary Data 1 and Supplementary Fig. 1).

**Sensitivity analysis of the model**. We performed a sensitivity analysis of our model to figure out those parameters, which upon changes of their numerical value have a large impact on the computed network states and thus deserve special care in the model parametrization procedure.

The sensitivity of stationary network states to changes of enzyme parameters was evaluated by means of π-elasticity coefficients, local response coefficients and global response coefficients (defined in Methods). As reference state we have chosen the steady-state that is adopted if the concentrations of all external metabolites are put to their 24 h mean values. All sensitivity measures are given in the Supplementary Data 4.

The distribution of π-elasticity coefficients (EC), quantifying the regulatory importance of an enzyme, reveals a balance between activating and inhibitory regulatory effects (Fig. 4a). Inspecting the occurrence of the four different parameter categories (see Table 1) reveals that large negative ECs are mainly accounted for by parameters of the category "N", which determine the deviation of the rate law from a hyperbolic shape (e.g., the exponent $n > 1$ in a Hill equation). The large group of parameters with EC = 1 is constituted by the $V_{max}$ values, which commonly occur as pre-factor of the rate law. The balanced share

of the category "KM" in the fraction of positive and negative ECs is due to the fact that increasing the Km value of a reaction substrate lowers the affinity and thus the reaction rate whereas increasing the Km value of a product (of a reversible reaction) has the opposite effect.

Infinitesimal response coefficients ($\widetilde{R}$) of all model parameters were computed with respect to 24 metabolic functions. For the statistical evaluation we recorded all parameters and associated metabolic functions meeting the condition $|\widetilde{R}| \geq R^*$ putting the threshold $R^*$ to 0.1, 0.5, and 1.0, respectively. 181 (=17.4%), 79 (=7.6%), and 33 (=3.1%) parameters affected at least one metabolic function (see Fig. 4b). Interestingly, there are some parameters with a strong impact on multiple metabolic functions (see Supplementary Fig. 5). Parameters affecting at least three different metabolic functions with $|\widetilde{R}| \geq 1$ are shown in Fig. 4d. They belong to processes located at the core of oxidative phosphorylation (F0F1-ATPase, complex I of the respiratory chain and mitochondrial uncoupling) or enzymes of the glycolytic pathway (glucokinase and phosphofructokinase 1).

Finite response coefficients ($R$) may differ from the infinitesimal ones owing to the non-linearity of rate laws. We calculated response coefficients for a finite parameter change by 50% and repeated the analysis shown in Fig. 4a, b (see Supplementary Figs. 3 and 4). A comparison with infinitesimal response coefficients (see Fig. 4c) shows, that the infinitesimal response coefficients have indeed the tendency to underestimate the system's response to finite parameter changes.

**Dynamic control analysis of liver metabolism**. The abundance of metabolic enzymes can be altered by regulated changes of protein expression and degradation, but also by chemical modifications (i.e., adduct formation with acetaldehyde) or inhibition by medical drugs. To check the functional consequences of such changes of enzyme abundances, we performed a dynamic metabolic control analysis (MCA), which differs from conventional MCA in that changes of the time course of model variables rather than changes of steady-state values are being calculated in response to a small perturbation of model parameters. To this end, we diminished the maximal activity of each of the 221 enzymes by 10% and computed the mean effect of this partial inhibition on the 24 h profile of 24 metabolic functions. This analysis revealed large differences in the capacity of individual enzymes to control metabolic functions by changes of their protein abundance (see Fig. 5). About 30% of all enzymes catalysing reactions that are close to the thermodynamic equilibrium have virtually no impact on the metabolic response of the liver. By contrast, about 10% of all enzymes exert significant control of more than a single metabolic function. As expected, inhibition of components of mitochondrial oxidative phosphorylation (e.g., reactions #38,39) influences almost all metabolic functions owing to the central function of ATP. But also less prominent reactions as the uptake of alanine (reaction #141) or the degradation of apo-protein B (reaction #156) have control over several metabolic functions. This kind of analysis can be used to identify potential targets for the treatment of hepatic diseases like steatosis. Such targets are indicated by red scores in column 17 of Fig. 5. Among the top-ranking putative anti-steatotic enzymes are GPAT (glycerol-P acyl transferase), mitochondrial uncoupling protein UCP or ACC1/2 (acetyl-CoA carboxylase), which are all discussed in the literature as potential targets for anti-steatotic drugs[15].

**Acute response of liver metabolism to a bolus of ethanol**. This example was chosen to illustrate how the model can be used to check the feasibility of hypotheses about the molecular basis of physiological or pathophysiological phenomena, in this case the

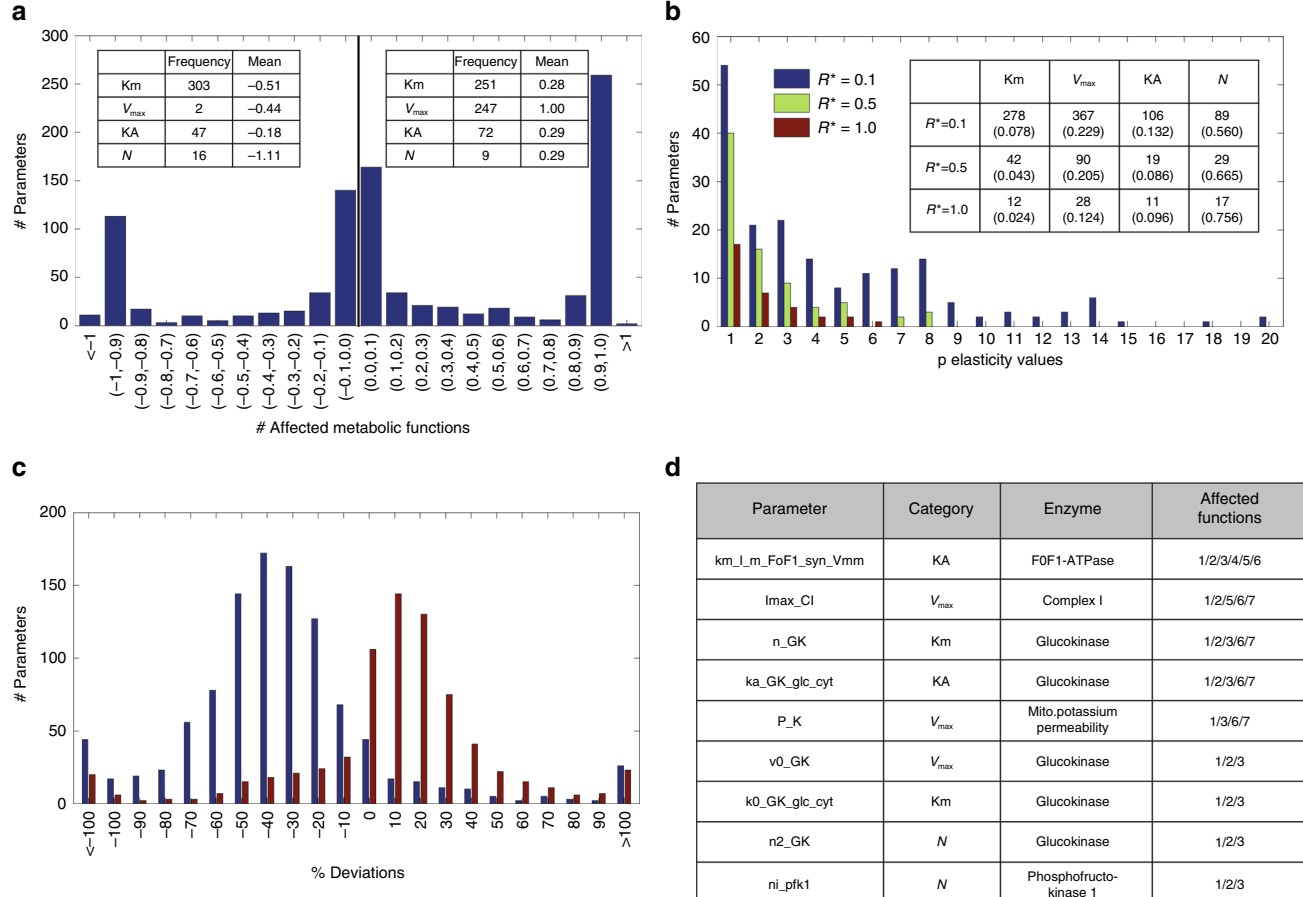

**Fig. 4** Parameter sensitivity analysis. **a** Distribution of π-elasticity coefficients (see equation (3) in Methods) and the relative occurrence of the four different parameter categories (see Table 1) and the category mean within the group of parameters with negative and positive ECs. Note that the total number of ECs (947) is smaller than the total number of parameters because in the reference state some reactions are not operative (e.g., fructose metabolism). **b** Frequency of parameters (y-axis) affecting a given number of metabolic functions (x-axis) with an infinitesimal response coefficient that is larger than the threshold value $R^*$ (0.1, 0.5, and 1.0). The inserted Table shows the absolute and relative frequencies (in brackets) of parameter categories. The relative frequency is given by the absolute frequency divided by the total frequency (see Table 1). **c** Relative differences $D = 100 \frac{\tilde{R}-R}{R}$ between infinitesimal response coefficients (with $R > 0.1$) and finite response coefficients computed for a 50% parameter change. Blue bars: 50% parameter reduction ($p \rightarrow p/2$). Red bars: 50% parameter increase ($p \rightarrow 1.5p$). **d** List of parameters affecting at least three different metabolic functions with an infinitesimal response coefficients $R \geq 1.0$. Numbering of metabolic functions: (1) glucose exchange rate, (2) lactate exchange rate, (3) pyruvate exchange rate, (4) glycerol exchange rate, (5) fatty acid uptake rate, (6) acetoacetate secretion rate, (7) β-hydroxybuterate secretion rate, (8) oxygen uptake rate, (9) ammonia uptake rate, (10) glutamine exchange rate, (11) glutamate exchange rate, (12) serine exchange rate, (13) alanine exchange rate, (14) urea secretion rate, (15) acetate exchange rate, (16) VLDL secretion rate, (17) glycogen storage, (18) cellular triglyceride concentration, (19) cholesterol synthesis lrate, (20) fatty acid synthesis rate, (21) mitochondrial membrane potential, (22) ATP/ADP ratio, (23) NAD/NADH ratio (cytosolic), and (24) NADP/NADPH ratio (cytosolic)

**Table 1 Model statistics**

| Model item | Number | Model item | Number |
|---|---|---|---|
| Enzymes & transporters | 209 | Kinetic parameters (total) | 1040 |
| - Catalyzing a single reaction | 189 | Parameters of type "KM" affinity constants of reactants | 604 |
| - Catalyzing several reactions | 20 | Parameters of type "KA" affinity constants of allosteric effectors | 139 |
| Compartments | 4 | Parameters of type "VMAX" maximal activities | 272[a] |
| Metabolites | 274 | Parameters of type "N" parameters determining the deviation from a hyperbolic rate law | 27 |
| Exchange fluxes with plasma | 21 | | |

[a]Note that there are more parameters of type "VMAX" than enzymes/transporters because 20 enzymes catalyze more than a single reaction

interrelation between alcohol drinking and development of a fatty liver. The main detoxification route of alcohol comprises the subsequent action of the enzymes alcohol dehydrogenase (ADH) and acetaldehyde dehydrogenase (ALDH) converting ethanol to acetate via the intermediate acetaldehyde (pathway #16, Fig. 1). Both reactions reduce $NAD^+$ to NADH. It is commonly argued that inhibition of fatty acid β-oxidation due to the lowered $NAD^+$/NADH ratio and the elevated de novo synthesis of fatty acids

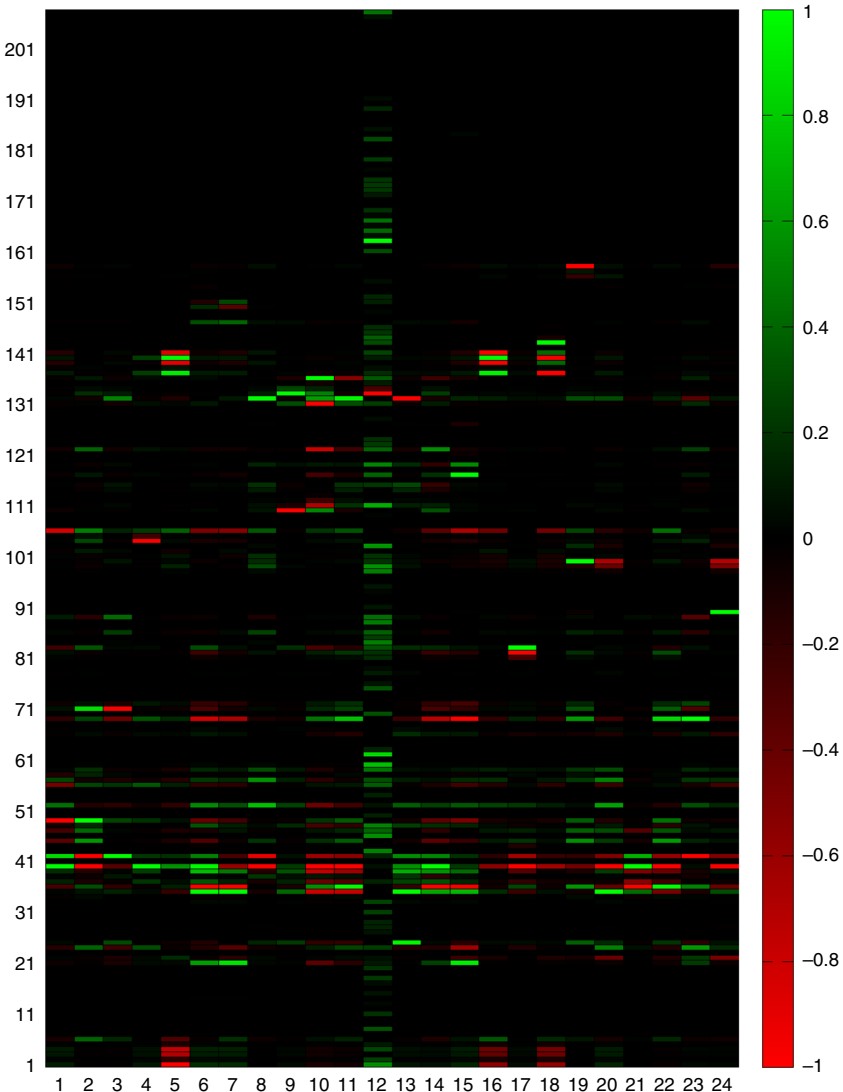

**Fig. 5** Dynamic control of 24 different metabolic functions. The maximal activity of each enzyme model was reduced by 10% and the resulting changes of the diurnal time-courses of 24 selected metabolic functions were quantified by means of the time-averaged response coefficient (defined by equation (6) in Methods). To quantify the relative impact of individual enzymes, the maximal positive and negative values means of the time-averaged response coefficients (Supplementary Data 2) were normalized to the range [−1, +1]. Numbering of metabolic functions: (1) glucose exchange rate, (2) lactate exchange rate, (3) pyruvate exchange rate, (4) glycerol exchange rate, (5) fatty acid uptake rate, (6) acetoacetate secretion rate, (7) β-hydroxybuterate secretion rate, (8) oxygen uptake rate, (9) ammonia uptake rate, (10) glutamine exchange rate, (11) glutamate exchange rate, (12) serine exchange rate, (13) alanine exchange rate (14) urea secretion rate; (15) acetate exchange rate, (16) VLDL secretion rate, (17) glycogen storage, (18) cellular triglyceride concentration, (19) cholesterol synthesis rate, (20) fatty acid synthesis rate, (21) mitochondrial membrane potential, (22) ATP/ADP ratio, (23) NAD/NADH ratio (cytosolic), (24) NADP/NADPH ratio (cytosolic)

from massively formed acetate are the key factors contributing to the development of a fatty liver (see e.g.,[16]). We used our model to check the soundness of this reasoning by simulating the effect of a single bolus of ethanol added to the standard plasma profile (see Fig. 6). A steep rise of the plasma alcohol level occurred if the ethanol infusion rate approached 100 μM/g/h (red curves) indicating saturation of ADH. The alcohol challenge induced a transient increase of the cellular triglyceride content (Fig. 6d) that was paralleled by an increased release of and lactate into the plasma (Fig. 6b). However, up to high plasma peak values of 38 mM (corresponding to 1.75 per mille) these transient alterations disappeared within a few hours after cessation of the ethanol bolus. The simulations reveal that lowered ratios of $NAD^+$/ NADH and pyruvate/lactate decrease the availability of pyruvate for the carboxylation to oxaloacetate and thus diminish the

formation of citrate by the citrate synthase. Hence, there is no significant citrate-dependent activation of the acetyl-CoA synthetase catalyzing the rate-limiting step of fatty acid synthesis. Notably, these model-based findings are in concordance with early experiments of Guynn[17] who observed a decrease of malonyl-CoA, citrate, and the activity of the acetyl-CoA carboxylase in response to acute ethanol administration. Hence, long-term changes in the expression of lipogenic enzymes remains the likely mechanism accounting for the development of an alcoholic fatty liver. Indeed, the transient increase of the toxic intermediate acetaldehyde (Fig. 6c) can activate the pro-lipogenic transcription factor SREB-1[18] and inhibit the transcription factor PPAR-α controlling the expression of enzymes involved in the β-oxidation of fatty acids[19].

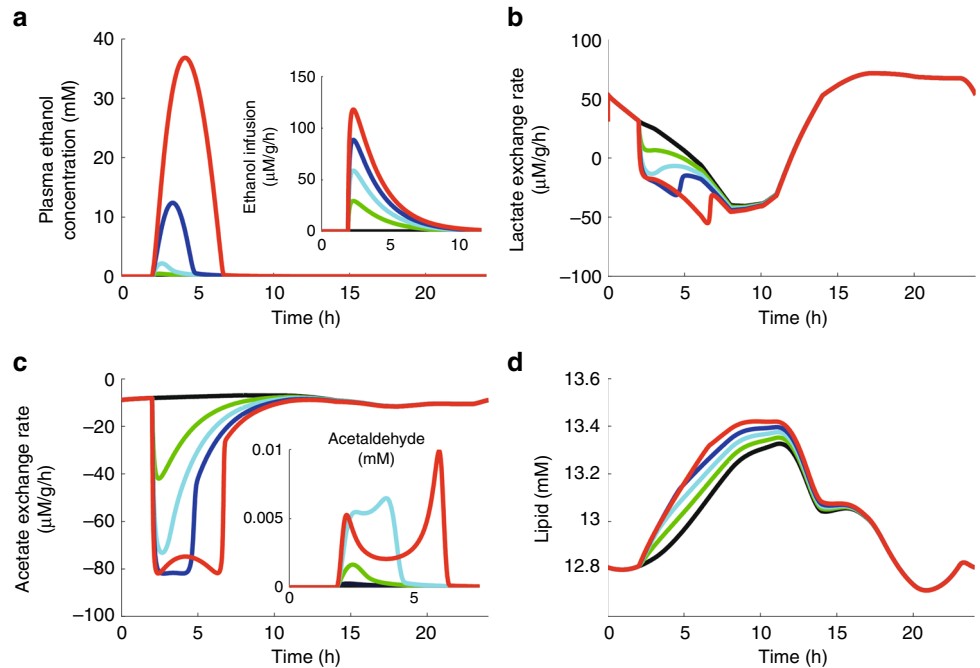

**Fig. 6** Response of liver metabolism to an alcohol challenge. **a** Plasma profile of ethanol at three different alcohol infusion profiles depicted in the small insertion. **b** Lactate exchange flux. **c** Acetate exchange flux. The profile of the toxic intermediate acetaldehyde is depicted in the small insertion. Note that the inhibition of the alcohol dehydrogenase at high ethanol concentrations accounts for the delayed steep rise of acetaldehyde if ethanol has already dropped to low level. **d** Triglyceride content. Whereas the plasma level of ethanol was normalized at times 5–7 h, the cellular increase of triglyceride persisted about 4 h up to time point $T = 11$ h

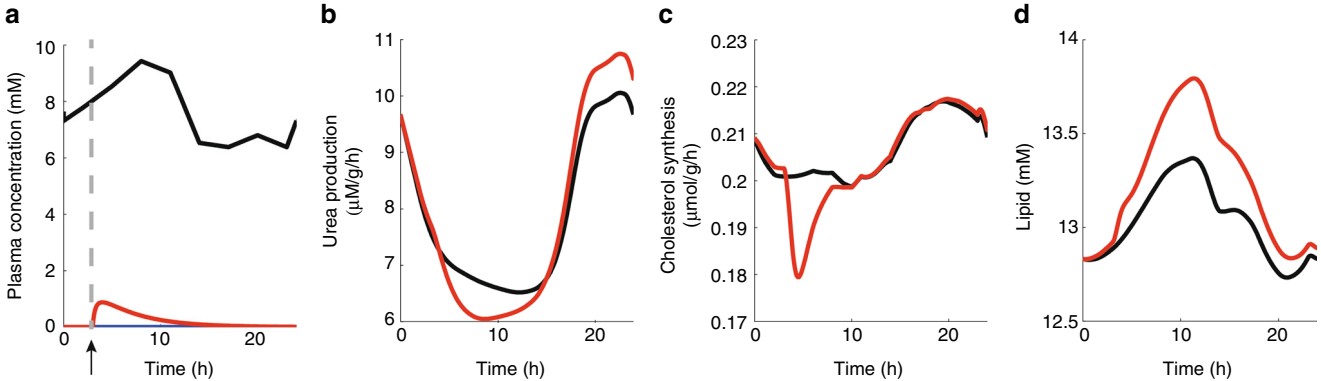

**Fig. 7** Metabolic effects of a single dose of valproate. **a** 24 h plasma glucose profile of the rat (black curve) and plasma profile of VPA after a bolus given at $t = 2$ h (red curve) adopted from[38] as the detoxification of VPA by the microsomal CYP system is not part of the model. **b** Urea production rate. **c** Synthesis rate of cholesterol. **d** Cellular triglyceride content. Black curves refer to the reference case in the absence of VPA

**Drug-induced steatosis by valproate**. An important application of the model consists in the quantitative investigation of metabolic changes elicited by medical drugs or toxins acting on specific metabolic enzymes. We have chosen the anticonvulsive drug valproic acid (VPA) as example. Clinical experience with VPA therapy has shown a number of fatal cases of hyperammonemia[20] and excessive triglyceride accumulation[21]. These adverse effects can be attributed to the VPA-dependent inhibition of two key enzymatic steps. Being chemically an analog of a medium-chain fatty acid, VPA is activated to valproyl-CoA thus sequestering CoA and acting as competitive inhibitor of the carnitin-palmitoyl-transferase 1 (CPT1)[22]. VPA also inhibits the N-acetyl-glutamate synthetase (AGS) and thus the formation of acetyl glutamate, a strong activator of the urea cycle[23]. The interactions of VPA with CPT1 and AGS have been kinetically characterized and thus could be included in the model in detail.

Fig. 7 shows the diurnal metabolic profile of selected metabolic functions in response to a single dose of VPA. At elevated plasma levels of VPA, the synthesis of urea is reduced (mean rate within 2–12 h = 7.87 μmol/g/h vs. 8.21 μmol/g/h for the reference case). However, across 1 day, the mean urea production rate is not significantly altered (9.56 vs. 9.44 μmol/g/h). In contrast, the triglyceride content is increased during the whole time span of VPA metabolization. Intriguingly, the simulation predicts a reduced synthesis rate of cholesterol. Such a cholesterol-depleting effect of VPA in the liver has not been reported so far but may account for a statistically significant drop of total serum cholesterol in epileptic outpatients on anticonvulsant monotherapy with VPA[24].

**Assessing the severity of inherited metabolic disorders**. The model offers the possibility to assess the severity of a hereditary

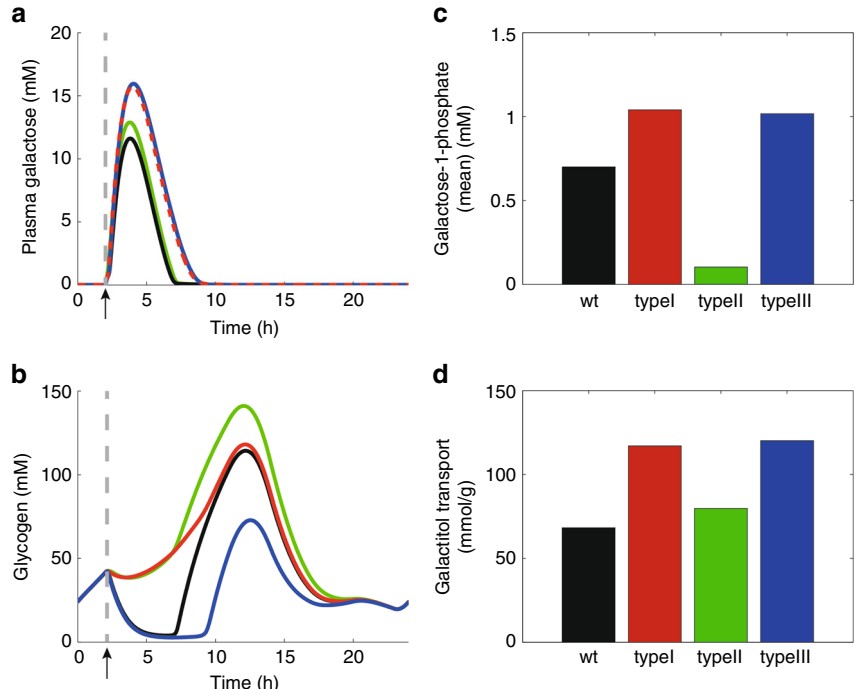

**Fig. 8** Dynamic changes in hepatic galactose metabolism. Rate equations for the enzymes of the Leloir pathway and kinetic parameters of the deficient enzymes (type I: GALT–red curves; type II: GALK–green curves; type III: GALE–blue curves) are given in Supplementary Note 4. The simulations were performed by starting at time $T = 2\,h$ the influx of galactose to the normal plasma. The influx was modeled by a phenomenological function taking into account the time delay between oral galactose intake and galactose appearance in the plasma. **a** Plasma profiles of galactose. **b** Cellular glycogen content. **c** Mean cellular Gal1P. **d** Mean galactitol transport rate

metabolic liver disease just knowing the altered kinetic properties of the defective enzyme. In this example, we focus on galactosemia, a rare genetic metabolic disorder that is caused by a deficiency of one of the three enzymes GALK, GALT, and GALE (cf. pathway #3, Fig. 1), which together constitute the so-called Leloir pathway converting galactose to UDP-glucose. Galactose can also be reduced to galactitol by the NADPH-dependent aldose reductase. Galactitol is an osmotically active compound that is considered a major factor in cataract formation if produced excessively. Finally, galactose can be oxidized to galactonate by the NAD-dependent galactose dehydrogenase. Galactonate is either released into the plasma or further metabolized to xylulose.

The clinical manifestations of each enzyme deficiency differ markedly. Patients with GALK deficiency (type II galactosemia) may present cataracts only. In contrast, GALT deficiency (type I classical galactosemia) is potentially lethal and demonstrates long-term, organ-specific complications[25]. While the molecular mechanisms underlying the pathogenesis of Type I galactosemia are still poorly understood, it is generally accepted that the intermediate galactose-1-phosphate (Gal1P) is the toxic metabolite responsible for the galactosemia phenotype.

Fig. 8 shows simulated temporal changes of the hepatic galactose metabolism elicited by a galactose challenge of a normal subject and of three patients suffering from galactosemia of type I, II, or III, respectively. Taking the area under the curves (AUC) of gal1P and galactitol as risk markers for systemic clinical complications and cataract, respectively, (Fig. 8c, d) the patient with GALT deficiency has by far the highest risk of systemic complications and an equally high risk of cataract than the patient with GALK deficiency. It has to be noted that this is a computational case study of individual patients. Note that kinetic alterations of genetic GALE variants are a continuum entailing a large scatter in the severity of galactosemia III[26].

**Metabolic phenotyping of liver tumors**. The gene expression profile of a tumor typically deviates strongly from that of the corresponding normal tissue[27] but the functional implications of these deviations remain often elusive. The model offers the opportunity to unravel the functional implications of changes in the protein abundance of metabolic enzymes in patient-specific liver tumors. To this end, we used the ratio of protein abundances of enzymes measured in the tumor cell and the normal hepatocyte by quantitative proteomics (Supplementary Data 3) to scale the corresponding ratio of $V_{max}$-values. As an example, Fig. 9 shows diurnal variations of selected metabolic functions of an adenoma and two hepatocellular carcinoma (HCC) resected from the liver of three individual patients (Supplementary Note 5). The simulations reveal remarkable differences between the metabolic profiles of the adenoma and the two HCC. Although the two HCC both display all metabolic features of the Warburg effect as a high rate of glucose consumption and lactate formation, depleted glycogen stores, reduced oxygen uptake, and lowered ATP/ADP ratio, the adenoma shows an almost normal glucose metabolism but severe alterations in the lipid metabolism (e.g., lowered rates of fatty acid uptake and TAG synthesis). The oxidative phosphorylation capacity is reduced in the two HCC but increased in the adenoma. Comparing metabolic profiles of the two HCC, there is a high similarity with respect to the glucose and energy metabolism but remarkable heterogeneity in the lipid and amino acid/nitrogen metabolism. For example, the uptake rates of glutamine, an important tumor substrate with several pleiotropic effects[28], differ by a factor of about five. Taken together, this pilot study points to the existence of a unique, patient-specific metabolic profile of tumors despite some basic metabolic communalities (e.g., Warburg effect in the two HCC).

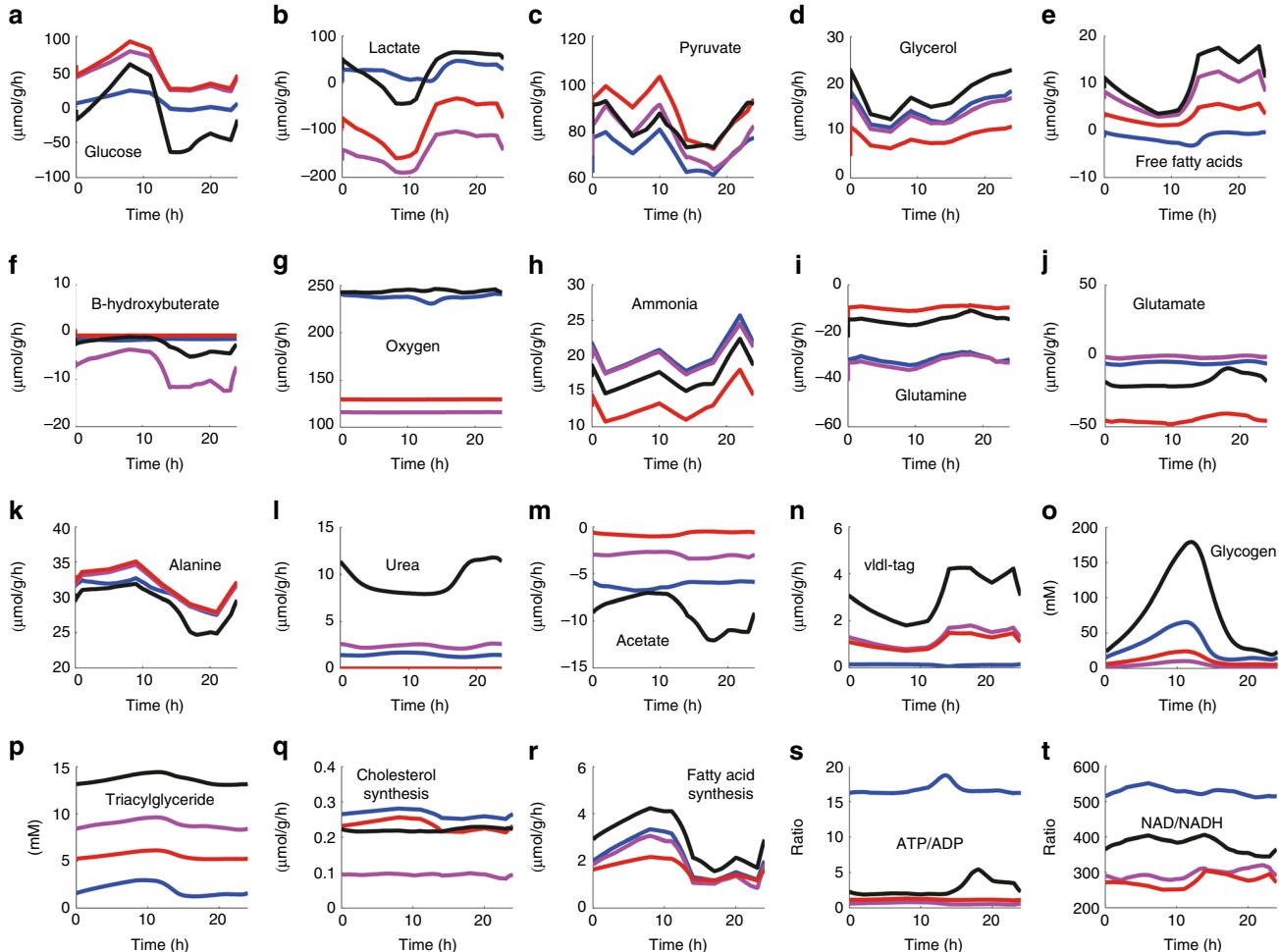

**Fig. 9** Metabolic profiling of liver tumors. Black lines: Normal hepatocyte. Red lines: T1 HCC, Pink lines: T3a HCC. Blue lines: Adenoma. The tumor tissues were obtained from three different patients and the proteome was determined by shot-gun proteomics. Relative protein abundances were used to scale maximal enzyme activities (Supplementary Data 3). The diurnal metabolite-hormone profile used as model input was the same as in Fig. 2. **a** glucose exchange rate, **b** lactate exchange rate, **c** glycogen content, **d** pyruvate exchange rate, **e** glycerol exchange rate, **f** oxygen uptake rate, **g** ATP/ADP ratio, **h** NAD/NADH ratio (cytosol); **i** acetate exchange rate, **j** free fatty acid exchange rate, **k** TAG content, **l** β-hydroxybutyrate exchange rate, **m** cholesterol synthesis, **n** fatty acid synthesis rate, **o** TAG export by VLDL, **p** ammonia exchange rate, **q** glutamine exchange rate, **r** glutamate exchange rate, **s** alanine exchange rate, **t** urea synthesis rate

## Discussion

Our work was inspired by both medical and methodological challenges. The bio-medical challenge was to establish a modeling platform that enables in silico studies of the response of the liver metabolism to variations of the external and internal conditions.

In order to illustrate the utility of the model for basic research and a variety of medical applications, we simulated temporal variations in the metabolic state of the liver in response to diurnal changes in the plasma profile of metabolites and hormones at normal conditions, nutritional challenges, drugs with metabolic side effects, enzymopathies, and altered gene expression profiles in liver tumors. The value of such model simulations consists in the ability to monitor simultaneously a large array of metabolites and fluxes in vivo. This is an exquisite situation, which can hardly be achieved in experiments because of ethical, technical, and economical restrictions. Hypotheses on molecular mechanisms underlying the system's behavior are commonly based on a restricted set of experimental observations of system variables and parameters. The presented model provides a means to check the feasibility of such hypothesis by filling the observational gaps by computation. The simulation of metabolic effects elicited by an alcohol bolus may serve as

example for such model-based hypothesis testing. Our simulations demonstrate the capability of liver metabolism to rapidly return to the normal state even after pronounced ethanol challenges, thus excluding alcohol-induced steatosis to result merely from metabolic dysregulation.

Applying the model to physiological or pathological states of the liver deviating from the reference state for a longer time period, adaptive alterations of enzyme abundances have to be taken into account. Hitherto, mechanistic models of the cellular protein turnover including the regulation of gene expression and proteolysis are not available yet. As an appropriate substitute, one may take advantage of the ongoing progress in quantitative proteomics by using experimentally determined changes of protein abundances to scale the maximal activities of enzymes and membrane transporters[6]. We used this approach to study the functional implications of altered enzyme levels in liver tumors. Our preliminary results suggest the existence of patient-specific metabolic profiles, which eventually may help to optimize individualized drug therapies. This computational approach may serve as an example for a proteome-based metabolic profiling of other liver diseases, such as liver steatosis, liver fibrosis, or liver cirrhosis.

The methodological challenge was to provide a paradigm for the feasibility of a large-scale kinetic modeling approach that takes into account the biochemistry of the system. Most common in the modeling of large metabolic networks is the application of constraint-based optimization methods, also known as flux-balance analysis (FBA)[29]. The applicability of this approach is restricted to the calculation of stationary flux distributions, which are obtained by optimizing a plausible objective function that relates basic cellular functions as, for example, the production rate of biomass or energy, to the fluxes through the constituting biochemical reactions. Additionally, upper flux constraints have to be imposed to assure a finite solution of the optimization problem. See ref.[30] for a review of the genome-wide metabolic flux distributions for various cell types based on constraint-based methods. For a more detailed review on current activities aiming at the development of mathematical whole-cell models we refer the reader to the report of the 2015 Whole-Cell Modeling Summer School[31].

Another modeling approach to large systems consists in the constructions of phenomenological kinetic models, where the need for detailed enzyme characterization is bypassed by using simplified rate laws, e.g., of the mass-action type or lin-log type. Such rate laws have fewer free scalable parameters that can be conveniently estimated by the ensemble modeling methods[32]. But naturally, this comes at the price of limited predictive power and ability to incorporate biochemical information. FBA models lack time responses and metabolite concentrations, while simplified rate laws lack the possibility to incorporate changes in kinetic parameters (as in inherited genetic diseases), enzyme isoforms (e.g., in cancer cells), and allosteric regulation.

The limitations of the above modeling techniques can be overcome by kinetic models resting at the principles of chemical kinetics and thermodynamics and including all (for the purpose of the model) relevant molecular details of enzyme regulation[33]. Hitherto, the development and validation of such biochemistry-based kinetic models has been limited to small metabolic subsystems comprising 10–30 reactions (see e.g., the model repositories[34,35]). Frequently quoted arguments for such a long standstill in the development of larger kinetic models are the lack of kinetic information on metabolic enzymes for most cell types and the general perception that data from enzymatic assays do not reflect the in vivo situation. The latter argument must be doubted by noting that all well-validated and often cited fit-for-purpose kinetic models of small metabolic systems—from the Heinrich-Rapoport models of red cell metabolism[36] to the model of the mammalian methionine cycle[37]—are ultimately based on in vitro enzyme parameters. There is simply no better information source for enzyme regulation than in vitro assays. Taking into account species-specific differences in kinetic constants and critically checking the plausibility of reported values by biochemical arguments (e.g., thermodynamic feasibility) may considerably reduce the degree of uncertainty in the choice of kinetic parameters. At the end, the decisive argument for the validity of the chosen numerical values of enzymatic parameters is the correct functioning of the network under a multitude of physiological conditions.

Finally, we want to emphasize that the development of a kinetic metabolic model of this complexity and detail was possible because metabolism is the by far best investigated cellular subsystem and, therefore, better accessible to mechanistic mathematical modeling than signaling or gene-regulatory networks. Our research was inspired by the vision that in the end all this information on individual enzymes can be brought together in large dynamic network models.

## Methods

**Model calibration.** We used a constraint optimization procedure to estimate numerical values for the $V_{max}$ values of all enzymes. For each of the 585 model simulations, we forced the concentration values of 170 internal metabolites $M_i$ ($i = 1,...,170$) to remain within the concentration ranges $[M_i^{min}, M_i^{max}]$ defined on the basis of in vivo and in vitro measurements (see Supplementary Data 1). This condition was implemented by introducing a penalty function H($M_i$) punishing calculated metabolite concentration falling out of the expected range:

$$H(M_i) = \begin{cases} (M_i - M_i^{max})^2 & \text{if } M_i > M_i^{max} \\ (M_i^{min} - M_i)^2 & \text{if } M_i < M_i^{min} \\ 0 & \text{else} \end{cases} \quad (1)$$

With this setting, the constrained optimization problem can be converted into an unconstrained optimization problem:

$$F = \sum_{\alpha=1}^{21} \left[ \left(\frac{1}{\bar{y}_\alpha^{exp}}\right)^2 \sum_{i=1}^{N_\alpha} (y_{\alpha i}^{exp} - y_{\alpha i}^{mod})^2 + \left(\frac{1}{\bar{M}^{exp}}\right)^2 \sum_{i=1}^{N_\alpha} \left(\sum_{j=1}^{M} H(M_{\alpha ij})\right) \right] \rightarrow \text{MINIMUM} \quad (2)$$

where the mean value of the observed variable in experiment ($\alpha$), $\bar{y}_\alpha^{exp} = (1/N_\alpha)\sum_{i=1}^{N_\alpha} y_{\alpha i}$, and the mean concentration value of all metabolites, $\bar{M}^{exp}$ ($\bar{M}^{exp} = 1.48$ mM), were used to deal with dimensionless variables and to properly scale the relative contributions of the two additive terms.

**Statistical measures.** For the sensitivity and control analysis of the model we used the following measures:
(i) The $\pi$-elasticity coefficient, which is defined as partial derivative of the rate $v$ of the isolated enzyme, i.e., at fixed values $\{E\}_0$ of the enzyme's effectors as reactants, allosteric effectors, and hormones with respect to an (infinitesimal) small perturbation of the parameter $p$,

$$\pi = \frac{p}{v}\frac{\partial v}{\partial p}\bigg|_{\{E_0\}} \quad (3)$$

(ii) The response coefficient, which is defined as change of the steady-state value of an arbitrary model variable $Y_i$ (e.g., metabolite concentration, reaction rate, and membrane potential) caused by a change of model parameter $p_j$,

$$R_{ij} = \frac{p_j}{\Delta p_j}\frac{\Delta Y_i}{Y_i} = \frac{1}{\lambda - 1}\left(\frac{Y_i(\lambda p)}{Y_i(p)} - 1\right) \quad (4)$$

whereby the second term on the right-hand side of relation (2) holds if the parameter change is expressed as $\lambda$-fold change of the initial parameter value, $p_j \rightarrow \lambda p_j$.

In the control theory of metabolism it is common to study the response of the system to infinitesimal parameter changes,

$$\bar{R}_{ij} = \lim_{\Delta p_j \rightarrow 0} \frac{p_j}{\Delta p_j}\frac{\Delta Y_i}{Y_i} = \lim_{\lambda \rightarrow 1}\frac{1}{\lambda - 1}\left(\frac{Y_i(\lambda p)}{Y_i(p)} - 1\right) = \frac{p_j}{Y_i}\frac{\partial Y_i}{\partial p_j} \quad (5)$$

We will refer to $R_{ij}$ and $\bar{R}_{ij}$ as finite and infinitesimal response coefficients.
(iii) For the sensitivity analysis of non-stationary states, we defined the time-averaged response coefficient

$$\langle R \rangle_T = \frac{\int_0^T |Y_{pert} - Y_{ref}| \, dt}{\int_0^T |Y_{ref}| \, dt} \quad (6)$$

where $Y_{ref}$ and $Y_{pert}$ denote the time-dependent changes of model variable $Y$ in the absence (ref) and presence (pert) of a perturbation and $T$ is the time interval of interest.

**Code availability.** An executable SBML file of the model is available from the authors on request.

**Data availability.** All data and public data sources used for the development, calibration, and exemplary model simulations are contained in the supplementary information.

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

## Acknowledgement

This work was funded by the German Systems Biology Programs "Virtual Liver", grant no. 0315741,"LiSyM", grant no. 31L0057 and 031L0058, the e:Bio (Module I) project "HepatomaSys", grant no.0316172A, all sponsored by the German Federal Ministry of Education and Research (BMBF), and the Max Planck Society.

## Author contributions

N.B. and H.-G.H. designed and guided the project and have written the manuscript. N.B. and S.B. have implemented the model. T.W. and M.S. provided the tumor samples and clinical data. D.M. performed the proteomics. M.K. contributed to the development of the signaling module and design of the figures. I.W. supported the literature search for kinetic data and revised the manuscript.

## Additional information

**Competing interests:** The authors declare no competing interests.

