## [Peer Review File · Nature Communications]

Reviewers' comments:

Reviewer #1 (Remarks to the Author):

This paper describes a very detailed kinetic model of hepatocyte metabolism. The model is reconstructed based on a thorough literature review and then evaluated for its ability to simulate various conditions. The model allows to simulate fluxes in the hepatocytes based on given inputs, e.g. metabolite concentrations in the plasma. The paper is clearly a significant advancement in our ability to model a very important cell type in humans. I only have a few comments that I suggest the authors address:

- 1) I suggest it is specified in more details, e.g. in a table, how many reactions and metabolites there are in the model. Also the number of parameters should be specified. Now this information is given scattered throughout the text.
- 2) It is also not clearly described how the kinetics of the many different reactions was specified. This is given in the supplementary information, but a brief summary would be good to have in the main text.
- 3) It is also not very clear how the boundaries of the model are defined.
- 4) In discussion of Fig 5 it says that synthesis of urea is reduced, but in the figure the urea production is given as negative values and in fact the rate is increasing. This should be clarified.

Reviewer #2 (Remarks to the Author):

In this paper, the authors develop a large scale, kinetic model of liver metabolism encompassing 209 reactions. The model seeks to capture known allosteric and hormone-mediated regulation of enzyme activity. The model is a bottom up model using in vitro data. Rate laws and all saturation and allosteric constants (945 parameters) are collated from the literature and treated as deterministic values.

The 209 maximum enzyme activities were fitted to "26" (actually 21) metabolic in vivo experiments comprising 585 "independent" measurements of exchange fluxes and metabolites. The fit was constrained by penalising any fit for having metabolite concentrations outside the reported range for 170 internal metabolites. The resultant model is used in several simulations to illustrate its utility.

The use of metabolite constraints when fitting the model is a highlight of the effort. It ensures that the fit produces a plausible and relatively robust model given the range of conditions encountered in the in vivo data.

The complete lack of validation of predictions is a major concern. Even where data were collected for the paper (tumor) only input (protein levels) were used; no metabolomics was performed.

Major issues

1. 945 parameters are picked from literature in vitro experiments and treated as deterministic. This approach is associated with numerous problems:
 - a. In vitro data can be very wrong as they do not reflect in vivo conditions; this results in both random and systematic errors in parameters
 - b. In vitro estimates from different groups are associated with very large errors; Brenda lists 41 values for the K_m for ATP in PFK2 and the range covers more than a magnitude (0.0151 – 0.25 mM)
 - c. Parameter fitting in enzymes is generally associated with very high cross correlation

coefficients.

d. The paper has no recognition let alone exploration of the possible effect of this source of stochasticity.

2. Diurnal Changes of Liver Metabolism

a. There is a surprising lack of validation. Hepatocytes are readily available and could be challenged short term with the input concentration with labelled compounds to determine output fluxes, concentrations and ratios.

b. Concentrations stay within range (S2) but this was of course part of the fitting process (the diurnal cycle data are not outside the training data).

3. Nutritional Challenges: Acute Response of Liver Metabolism to a Bolus of Ethanol

a. Again, the argument is built around the model being correct without any experimental validation of the conclusions.

b. Models are ideal for producing non-obvious hypotheses, but without experimental testing there is no validation or attempted falsification performed.

4. Drug Effects on Metabolic Enzymes. The urea production is lower (10 h) and higher (20 h); is it really lower overall?

5. Inherited Disorders of Metabolic Enzymes: Galactosemia. This is a self-evident test. Obviously, a GalK KO will accumulate less Gal1P, while the other two will accumulate Gal1P. No large scale kinetic model is required to explain this. Arguably, it would be more relevant to explore why Type 1 and Type 3 differs clinically, when they are very similar in the simulation.

6. Metabolic Phenotyping of Liver Tumors

a. Again, there is a surprising lack of validation. It would be natural to use measured enzyme concentration and key plasma metabolites to predict concentrations and fluxes and compare against actual intracellular concentrations (fluxes would be more difficult).

b. There is no account of the significant change in the dominant enzyme isoforms happening during tumorigenesis. Different isoforms have very different regulation, so it is unclear what the model is actually predicting or if the model even consider the amount of the alternative isoforms.

7. The introduction and discussion are very brief on discussing previous and parallel efforts on large scale kinetic models.

8. The others fail to make the model available for the readers to test and use. This greatly reduce any utility that this work might have.

Minor issues

1. Page 6: There appear to be 21 not 26 metabolic experiments used for the fit (based on Table 1, fitting function and the Supp)

2. Page 7 Note: Min F does not minimize distance between 100,000 model values; it minimises the distance to 585 values, while constraining concentration.

3. Supplement 4 is clearly not a model validation since it is used to fit the model.

4. Diurnal Changes of Liver Metabolism: nad/nadh ratio of 400. Please clarify that this is the free cytosolic NAD/NADH pool.

Point-by-point response

General remark

In the revised manuscript, all requests and comments of the reviewers regarding the precise outline of model details have been addressed. Regarding the further validation experiments as requested by reviewer #2, we attach a *ready-for-submission* manuscript of a follow-up study in which we have used the model to characterize the metabolic phenotype of hepatocellular hepatoma (HCC) in mice, including extensive experimental validation of model predictions. Merging the two manuscripts into one single publication appears to be not appropriate, just because of the additional space required for an adequate outline of all experiments. However, we would be pleased if NATURE COMMUNICATIONS would consider the option to publish these two closely related manuscripts side by side.

In the following we provide a point-by-point response to the comments of the reviewers. Changes in the manuscript are indicated in bold blue.

Reviewer #1 (Remarks to the Author)

1) I suggest it is specified in more details, e.g. in a table, how many reactions and metabolites there are in the model. Also the number of parameters should be specified. Now this information is given scattered throughout the text.

An additional table was added specifying the number of reactions, transport processes, enzymes, metabolites, kinetic parameters.

model item	number
Enzymes & Transporters	209
Enzymes catalyzing a single reaction	189
Enzymes catalyzing two or more reactions	20
Compartments	4
Metabolites	274
Exchange fluxes with the extra-cellular space	21
Kinetic parameters	945

2) It is also not clearly described how the kinetics of the many different reactions was specified. This is given in the supplementary information, but a brief summary would be good to have in the main text.

We added a short section that describes our general strategy to specify the rate equations:

The mathematical form of the kinetic rate law for enzymes and membrane transporters is dictated by the reaction mechanism and was taken from enzymological *in vitro* studies, preferably for the liver of the rat or, if not available, in the species order mice → human → bovine → dog. The same ranking of species was applied for the retrieval of numerical values for the kinetic parameters. For every kinetic parameter we cited one experimental reference. Most often several numerical values for the same kinetic parameter and the same species were reported. In such case we used one representative value that fits with the majority of the reported values and that - whenever possible - was taken from an enzyme assay that reported consistent values for other kinetic parameters. Mathematical terms in the

rate laws related to allosteric enzyme effectors which are not included in the model were neglected. Their average contribution is thus indirectly contained in the fitted Vmax values.

3) It is also not very clear how the boundaries of the model are defined.

To clarify the model boundaries we have added this explanation to the main text:

Model boundaries The model boundaries are given by the metabolite and hormone concentrations in the extracellular space. The flux through reactions not considered in the model are put to zero, i.e. carbon influx into and efflux from the modelled metabolic subsystem is exclusively mediated by the exchange fluxes through the plasma membrane.

4) In discussion of Fig 5 it says that synthesis of urea is reduced, but in the figure the urea production is given as negative values and in fact the rate is increasing. This should be clarified.

The simulation of urea synthesis in response to a single VPA bolus was not correctly described. We replaced the respective passage by the following:

During elevated plasma levels of VPA, the synthesis of urea is reduced (mean rate within 2-12h = 7.87 $\mu\text{mol/g/h}$ versus 8.21 $\mu\text{mol/g/h}$ for the reference case). However, across one day the mean urea production rate is not significantly altered (9.56 $\mu\text{mol/g/h}$ versus 9.44 $\mu\text{mol/g/h}$). In contrast, the triglyceride content is increased during the whole time span of VPA metabolization.

Reviewer #2 (Remarks to the Author):

The complete lack of validation of predictions is a major concern. Even where data were collected for the paper (tumor) only input (protein levels) were used; no metabolomics was performed.

The criticism of a "complete" lack of validation of predictions cannot be accepted. The simulations of each showcase (alcohol metabolism, enzyme defects, cancer metabolism), recapitulate reported metabolic features of the liver (that we refer with references) and thus lends further support to the reliability of the model. However, a further in-depth experimental validation of the suggested molecular mechanisms accounting for the newly predicted features requires considerable experimental effort and thus must be reserved to subsequent studies.

Major issues

1. 945 parameters are picked from literature in vitro experiments and treated as deterministic. This approach is associated with numerous problems:

a. In vitro data can be very wrong as they do not reflect in vivo conditions; this results in both random and systematic errors in parameters

b. In vitro estimates from different groups are associated with very large errors; Brenda lists 41 values for the K_m for ATP in PFK2 and the range covers more than a magnitude (0.0151 – 0.25 mM)

We are certainly aware of all these difficulties. In the Discussion, we added the following passage related to the problem of uncertain in vitro parameters:

Frequently quoted arguments for such a long standstill (in the development of larger kinetic models) are the lack of reliable kinetic information on metabolic enzymes for most cell types and the general perception that data from enzymatic assays do not reflect the in vivo situation. The latter argument must be doubted by noting that all well-validated and often cited fit-for-purpose kinetic models of small metabolic systems - from the Heinrich-Rapoport models of red cell metabolism to the model of the mammalian methionine cycle - are ultimately based on in vitro enzyme parameters.

There is simply no better information source for enzyme regulation than in vitro assays. Taking into account species-specific differences in kinetic constants and critically checking the plausibility of reported values by biochemical arguments (e.g. thermodynamic feasibility) instead of blindly grasping parameter values from public enzyme data bases may considerably reduce the degree of uncertainty in the choice of kinetic parameters. At the end, the most important argument for the validity of the chosen numerical values of enzymatic parameters is the correct functioning of the network under a multitude of physiological conditions.

Regarding the specific example of the PFK2 mentioned by the reviewer:

Brenda lists all values irrespective of species or source tissue. Whenever possible we took data reported for rat liver. If not available, we used data for mouse or human liver. The K_m for ATP in PFK2 given in Brenda shows a high homogeneity between 0.1 -0.3 mM. We chose 0.25 as the parameter that is in concordance with the majority of the reported values. Moreover, the source publication reports a number of further kinetic parameters which allows to create an experimentally consistent data base with a limited number of different data sources. In this particular case, choosing any value between 0.1 or 0.3 does not affect the model behavior at all as the cytosolic cellular ATP level of ~ 3mM is high enough to keep the enzyme saturated with ATP.

c. Parameter fitting in enzymes is generally associated with very high cross correlation coefficients.

From the systems modeling perspective this undeniable fact is even an advantage because possible errors of numerical values for binding constants are partially compensated by an appropriate choice of the maximal activity (the only free parameters in model calibration).

d. The paper has no recognition let alone exploration of the possible effect of this source of stochasticity.

The maximal enzyme activity (V_{max}) is the enzyme parameter having the most significant impact on the systems behavior. We checked the consequence of stochastic variations of this parameter (see grey-shaded ranges in Fig. 3).

2. Diurnal Changes of Liver Metabolism

a. There is a surprising lack of validation. Hepatocytes are readily available and could be challenged short term with the input concentration with labelled compounds to determine output fluxes, concentrations and ratios.

In principle yes, but

(i) isolated hepatocytes are (unfortunately) metabolically unstable, disconnected from the tissue and from cytokines sent from the endothelium of the sinusoids they continuously change their metabolic state.

(ii) temporal variation of external metabolites (more than 20 including glucose and fatty acids) together with related variations of hormones (glucagon and insulin) in one assay is impossible to implement so far

b. Concentrations stay within range (S2) but this was of course part of the fitting process (the diurnal cycle data are not outside the training data).

This conclusion of the reviewer is not correct. In the training set, only few distinct parameters were varied and hormone levels were either zero or set constant. In the diurnal cycle data, numerous exchange metabolites and hormone levels are changing simultaneously and still the internal metabolites remain within the expected concentration range.

3. Nutritional Challenges: Acute Response of Liver Metabolism to a Bolus of Ethanol

a. Again, the argument is built around the model being correct without any experimental validation of the conclusions.

b. Models are ideal for producing non-obvious hypotheses, but without experimental testing there is no validation or attempted falsification performed.

We provide a tool to assess very different questions around liver metabolism as for example the role of alcohol abuse for fatty liver. We show that excessive alcohol intake indeed leads to fat accumulation in the liver and that this process is reversible. The findings regarding the metabolic alterations during alcohol uptake are backed up by numerous experimental findings (which we cite). Our conclusion that alterations in gene expression should significantly contribute to the metabolic alterations observed in alcoholic liver disease is also in concordance with experimental findings (e.g. Gene Expression Profiling of

Alcoholic Liver Disease in the Baboon and Human Liver (**Papio hamadryas**). An in-depths study of the very important medical issue of the alcohol-induced fatty liver cannot be the aim of a showcase.

4. Drug Effects on Metabolic Enzymes. The urea production is lower (10 h) and higher (20 h); is it really lower overall?

The simulation of urea synthesis in response to a single VPA bolus was not correctly described. We replaced the respective passage by the following:

During elevated plasma levels of VPA, the synthesis of urea is reduced (mean rate within 2-12h = 7.87 $\mu\text{mol/g/h}$ versus 8.21 $\mu\text{mol/g/h}$ for the reference case). However, across one day the mean urea production rate is not significantly altered (9.56 $\mu\text{mol/g/h}$ versus 9.44 $\mu\text{mol/g/h}$). In contrast, the triglyceride content is increased during the whole time span of VPA metabolism.

5. Inherited Disorders of Metabolic Enzymes: Galactosemia. This is a self-evident test. Obviously, a GalK KO will accumulate less Gal1P, while the other two will accumulate Gal1P. No large scale kinetic model is required to explain this. Arguably, it would be more relevant to explore why Type 1 and Type 3 differs clinically, when they are very similar in the simulation.

The aim of this show case is to demonstrate that the model can make patient-specific predictions of the severity of a metabolic disorder if the kinetic parameters of the disease-causing deficient enzyme (obtainable in an enzyme assay) are known.

The severe form of galactosemia III has symptoms similar to those of galactosemia as suggested by our simulations for one particular case. The defective enzyme GALE is responsible not only for the interconversion of UDPgalactose and UDP-glucose but also UDP-N-acetylgalactosamine and UDP-N-acetylglucosamine. These molecules are key precursors for the synthesis of sugar moieties in glycoproteins. It is likely that perturbations of the amino-sugar metabolism may be a causative factor in type III galactosemia III pathology. This is a motivation for us to incorporate the synthesis of amino sugars into an extended version of our model

6. Metabolic Phenotyping of Liver Tumors

a. Again, there is a surprising lack of validation. It would be natural to use measured enzyme concentration and key plasma metabolites to predict concentrations and fluxes and compare against actual intracellular concentrations (fluxes would be more difficult).

b. There is no account of the significant change in the dominant enzyme isoforms happening during tumorigenesis. Different isoforms have very different regulation, so it is unclear what the model is actually predicting or if the model even consider the amount of the alternative isoforms.

Together with experimental partners and clinicians, we have performed a follow-up study of the metabolic phenotype of a liver tumor in mice exhibiting the characteristics of the hepatocellular melanoma (HCC). We used exactly the same kinetic model as the one presented in this manuscript to infer tumor-specific changes of pathway fluxes and metabolites from changes in the abundance of

metabolic enzymes (determined by mass spectrometry) including tumor-specific isoforms and their specific regulatory properties. The model predictions underwent a thorough experimental validation. We enclose a copy of the manuscript of this combined modelling-experiment study as private and confidential information for the reviewers and the editor in order to further substantiate the physiological significance of the model. N. Berndt, the first author of this manuscript is also the first author of the follow-up paper.

7. The introduction and discussion are very brief on discussing previous and parallel efforts on large scale kinetic models.

In addition to the already cited large-scale kinetic models of Karr et al. and Khodayari et al., we are aware of only one further study by which we have included . We also added the following passage to the Discussion:

The difficulties in developing physiology-based large-scale metabolic models have prompted the emergence of alternative modelling approaches. Most common in the metabolic modeling of whole cells is the complete avoidance of kinetic models in favor of constraint-based optimization methods, also known as flux-balance analysis 35. The applicability of this approach is restricted to the calculation of stationary flux distributions which are obtained by optimizing a plausible objective function that relates basic cellular functions as, for example, the production rate of biomass or energy, to the fluxes through the underlying biochemical reactions. Additionally, upper flux constraints have to be imposed to assure a finite solution of the optimization problem. See 36 for a review of the genome-wide metabolic flux distributions for various cell types based on constraint-based methods. For a more detailed review on current activities aiming at the development of mathematical whole-cell models we refer the reader to the report of the 2015 Whole-Cell Modeling Summer School 37. The report outlines a 5-step pipeline for the development of whole-cell mathematical models based on genomic information 37. One such alternative consists in the constructions of phenomenological kinetic models where the need for detailed enzyme characterization is bypassed by using simplified rate laws, e.g. of the mass-action type or lin-log type. Such rate laws have fewer free scalable parameters that can be conveniently estimated by the ensemble modeling methods 32. This modeling approach has been applied to construct stationary flux distributions for whole cells like erythrocytes 33 or Escherichia coli 34. The obvious advantage of such models lies in the limited number of data needed for their implementation. But naturally this comes at the price of limited predictive power and ability to incorporate biochemical information. FBA model lack time responses and metabolite concentrations, while simplified rate laws lack the possibility to incorporate changes in kinetic parameters (as in inherited genetic diseases), isoforms (e.g. in cancer cells) and allosteric regulation.

8. The others fail to make the model available for the readers to test and use. This greatly reduce any utility that this work might have.

We generated a SBML model that allows usage of the model at any platform supporting the SBML standard. We attach this file to the submission of the revised manuscript

Minor issues

1. Page 6: There appear to be 21 not 26 metabolic experiments used for the fit (based on Table 1, fitting function and the Supp)

Actually, we used data from 25 different experiments (differing in the pair of input and output variables) testing 21 different metabolic features of the liver. In supplement 4, Figures 5, 9 and 12 show experimental and simulated data for several independent experiments. We changed the passage accordingly:

The $p = 209$ unknown V_{max} values were estimated by fitting the model to 585 measurements y_{ai} of exchange fluxes and internal metabolites obtained in 25 different experiments differing in the pair of input and output variables and covering 21 important liver functions (see Table 1).

2. Page 7 Note: Min F does not minimize distance between 100,000 model values; it minimises the distance to 585 values, while constraining concentration.

Absolutely correct. As the fitting procedure is sufficiently explained, we canceled this sentence.

3. Supplement 4 is clearly not a model validation since it is used to fit the model.

The supplement 4 was renamed to "**model calibration**".

4. Diurnal Changes of Liver Metabolism: nad/nadh ratio of 400. Please clarify that this is the free cytosolic NAD/NADH pool.

We added to the legend of Fig. 4:

Note that panel 23 shows the free cytosolic NAD/NADH ratio.

Reviewers' comments:

Reviewer #1 (Remarks to the Author):

I am satisfied with the revision and I think this paper is excellent. I have no further comments for improvement of the paper. I also think the authors have addressed all reviewer comments satisfactory.

Reviewer #3 (Remarks to the Author):

There is certainly much potential value in large-scale kinetic models of metabolism and liver metabolism is a prime target for the value of such computational models to understand mechanisms and to predict potential therapeutic interventions.

I have a couple concerns with the manuscript.

1. A sensitivity analysis of the model predictions to variation in the kinetic parameters would instill much more confidence in the validity of the arguments.
2. In the previous round of reviews, there was an exchange about fitting kinetic parameters and validity of parameters from in vitro assays among other things. I agree with the criticisms and such problems can be amplified with even larger kinetic models. A key way to mitigate this concern is as stated in #1.
3. The paper discusses the model development and then goes through a subsequent list of several different applications for which there are a lot of simulation data presented and some literature support to the predictions. The paper would benefit from figures that are a more targeted and concise delineation of the key prediction and key literature validation to support the prediction. Then the paper would benefit from a more explicit presentation of the "value added" from the model, addressing a question like "with this model, how can we understand which enzyme deficiency resulted in this phenotype which is not obvious otherwise". As it currently stands, the manuscript has several complex dynamic profiles from the simulation data and the reader gets lost in what you're supposed to be looking at.
4. Perhaps seemingly of lower consequence, but which I think is nonetheless very important...the tone is quite inappropriate. Multiple uses of phrases like "undoubtedly" or "blindly grasping" or "most reliable" in reference to principles that support their own modeling perspectives and approaches. This language totally shapes the perspective of the reader in an inappropriate way.

Point-by-point response to the referees' comments

We thank the reviewer for her/his suggestions for improvements of the manuscript which may increase the confidence in the validity of the model and which better highlight the extra value that the model may add to existing knowledge.

In the following we provide a point-by-point reply to the reviewers comments. All changes in the manuscript are marked in blue lettering.

Reviewer:

- 1. A sensitivity analysis of the model predictions to variation in the kinetic parameters would instill much more confidence in the validity of the arguments.*
- 2. In the previous round of reviews, there was an exchange about fitting kinetic parameters and validity of parameters from in vitro assays among other things. I agree with the criticisms and such problems can be amplified with even larger kinetic models. A key way to mitigate this concern is as stated in #1.*

Authors reply to points 1. and 2.

As suggested, we performed a detailed parameter sensitivity analysis of the model consisting of three parts: (1) Computation of π -elasticity coefficients which characterize the sensitivity of the isolated enzyme (with all non-parametric variables kept at fixed values) to small parameter changes, (2) computation of infinitesimal response coefficients quantitating the impact of small parameter changes on functional fluxes of the network and (3) computation of finite response coefficients quantitating the impact of larger parameter changes (+/- 50% of the initial value) on functional fluxes of the network. We also analyzed the difference between infinitesimal and finite response coefficients. The results of the sensitivity analysis are outlined in the new chapter "Parameter Sensitivity Analysis of the Model", in the new supplement S6 "Sensitivity Analysis" and in the supplementary Excel file "Sensitivity Analysis. XLS" which contains of computed sensitivity measures.

Reviewer:

- 3. The paper discusses the model development and then goes through a subsequent list of several different applications for which there are a lot of simulation data presented and some literature support to the predictions. The paper would benefit from figures that are a more targeted and concise delineation of the key prediction and key literature validation to support the prediction. Then the paper would benefit from a more explicit presentation of the "value added" from the model, addressing a question like "with this model, how can we understand which enzyme deficiency resulted in this phenotype which is not obvious otherwise". As it currently stands, the manuscript has several complex dynamic profiles from the simulation data and the reader gets lost in what you're supposed to be looking at.*

Authors reply to 3.

We replaced Figure 3 (which showed the 24h profile of 24 metabolic functions but without guiding the reader through the numerous small panels) by a new figure that delineates the diurnal profile of only a sub-set of metabolic functions related to the glucose- and lipid metabolism of the liver. The interrelations between these profiles are commented and substantiated by literature findings. We also added a new Table 3 to introduce the 24 metabolic functions to which we refer at a number of different places in the manuscript.

Furthermore, we revised the text of the showcases (alcohol-induced lipogenesis, drug effects, galactosemia, cancer metabolism) by explaining more thoroughly the simulation results shown in the respective Figs. 7-10. For each showcase we explicitly mention the extra value that the model adds to existing knowledge.

Reviewer:

4. Perhaps seemingly of lower consequence, but which I think is nonetheless very important...the tone is quite inappropriate. Multiple uses of phrases like "undoubtedly" or "blindly grasping" or "most reliable" in reference to principles that support their own modeling perspectives and approaches. This language totally shapes the perspective of the reader in an inappropriate way.

Authors reply to 4.

We omitted all phrases which could give the impression that we hold alternative modeling approaches to low esteem. Phrases as such criticized by the reviewer were thought as an “outcry” to remind the community of modelers that kinetic modeling has its own value beside other approaches that currently dominate the field. Obviously we shot beyond the target

REVIEWERS' COMMENTS:

Reviewer #3 (Remarks to the Author):

The authors satisfactorily addressed my previous critiques.

Please note there were several spelling and grammar issues in this revised version.

REVIEWERS' COMMENTS:

Reviewer #3 (Remarks to the Author):

The authors satisfactorily addressed my previous critiques.

Please note there were several spelling and grammar issues in this revised version.

Authors reply

The final document was checked for spelling and grammar issues by a native speaking scientist.